# Heart Rate from Progressive Volitional Cycling Test Is Associated with Endothelial Dysfunction Outcomes in Hypertensive Chilean Adults

**DOI:** 10.3390/ijerph20054236

**Published:** 2023-02-27

**Authors:** Cristian Alvarez, Marcelo Tuesta, Álvaro Reyes, Francisco Guede-Rojas, Luis Peñailillo, Igor Cigarroa, Jaime Vásquez-Gómez, Johnattan Cano-Montoya, Cristóbal Durán-Marín, Oscar Rojas-Paz, Héctor Márquez, Mikel Izquierdo, Pedro Delgado-Floody

**Affiliations:** 1Exercise and Rehabilitation Sciences Institute, School of Physical Therapy, Faculty of Rehabilitation Sciences, Universidad Andres Bello, Santiago 7591538, Chile; 2Escuela de Kinesiología, Facultad de Salud, Universidad Santo Tomás, Los Ángeles 4440000, Chile; 3Centro de Investigación de Estudios Avanzados del Maule (CIEAM), Universidad Católica del Maule, Talca 3460000, Chile; 4Laboratorio de Rendimiento Humano, Universidad Católica del Maule, Talca 3460000, Chile; 5Escuela de Kinesiología, Facultad de Odontología y Ciencias de La Rehabilitación, Universidad San Sebastián, Valdivia 8420524, Chile; 6Physical Therapy, Faculty of Rehabilitation Sciences Carrera de Kinesiología, Universidad Andres Bello, Concepción 4260000, Chile; 7Navarrabiomed, Hospital Universitario de Navarra (HUN), Universidad Pública de Navarra (UPNA), IdiSNA, 31006 Pamplona, Spain; 8CIBER of Frailty and Healthy Aging (CIBERFES), Instituto de Salud Carlos III, 28220 Madrid, Spain; 9Department of Physical Education, Sports and Recreation, Universidad de La Frontera, Temuco 4811230, Chile

**Keywords:** flow-mediated dilation, pulse wave velocity, carotid-intima media thickness, endothelial dysfunction, arterial hypertension, exercise test, vasculature

## Abstract

Background: A progressive volitional cycling test is useful in determining exercise prescription in populations with cardiovascular and metabolic diseases. However, little is known about the association between heart rate during this test and endothelial dysfunction (EDys) parameters in hypertensive (HTN) patients. Objective: To investigate the association between EDys markers (flow-mediated dilation [FMD], pulse wave velocity of the brachial artery [PWVba], and carotid-intima media thickness [cIMT]) and heart rate during a cycling test in HTN adults. A secondary aim was to characterize cardiovascular, anthropometric, and body composition outcomes in this population. Methods: This was a descriptive clinical study in which adults (men and women) were assigned to one of three groups: HTN, elevated blood pressure (Ele), or a normotensive control group (CG), and completed a progressive cycling test. The primary outcomes were FMD, PWVba, cIMT, and heart rate (HR) at 25–50 watts (HR_25–50_), 50–100 watts (HR_50–100_), and 75–150 watts (HR_75–150_) of the Astrand test. Secondary outcomes included body mass index (BMI), waist circumference, body fat percentage (BF%), skeletal muscle mass (SMM), resting metabolic rate (RMR), and estimated body age, as measured by a bio-impedance digital scale. Results: Analyses of the associations between FMD, PWV, and HR_25–50_, HR_50–100_, and HR_75–150_ watts revealed no significant association in the HTN, Ele, and CG groups. However, a significant association was found between cIMT and HR_75–150_ watts in the HTN group (R^2^ 47.1, β −0.650, *p* = 0.038). There was also a significant trend (*p* = 0.047) towards increasing PWVba in the CG, Ele, and HTN groups. Conclusion: Heart rate during a progressive cycling test is associated with the EDys parameters cIMT in HTN patients, with particularly strong predictive capacity for vascular parameters in the second and third stages of the Astrand exercise test compared to normotensive control.

## 1. Introduction

Atherosclerosis is a chronic disease characterized by the accumulation of lipoproteins in the inner layer of artery walls. This accumulation is often due to oxidative damage to low-density lipoprotein (LDL-c) [1]. The accumulation of LDL-c can lead to inflammation in the major arteries (e.g., carotid and brachial arteries), which typically progress to fibroatheromas [2]. However, before the development of atherosclerosis, an endothelial dysfunction (EDys) state is usually found. EDys is a phenotypic condition that is an intermediate pathology, characterized by a pro-thrombotic and pro-inflammatory state. This is the result of an imbalance between the actions of vasodilators and vasoconstrictors, which modifies the “function” and “structure” of the vasculature [3]. Traditional methods for detecting EDys are highly invasive, expensive, and time-consuming, such as coronary epicardial vasoreactivity and venous occlusion plethysmography. Therefore, non-invasive methods based on ultrasound imaging have been rapidly implemented in clinical management [4,5].

EDys is often associated with several health conditions, including arterial hypertension (HTN), obesity, coronary artery disease, chronic heart failure, peripheral artery disease, diabetes, metabolic syndrome, non-alcoholic fatty liver disease, and chronic renal failure [6]. In Chile, 26.9% of adults have HTN, and this prevalence is highly superior in older adults [7]. Therefore, it is estimated that an important number of adults and older adults will develop EDys, which will progress to atherosclerosis, or to an atheromatous plaque that, in turn, will increase the risk of stroke. Regarding “functional” parameters, the percentage of flow-mediated dilation (FMD) has been a more suitable and strong marker of vascular health in adults. Low values of FMD (i.e., <6.5%) denote an impaired vascular function associated with cardiometabolic risk [4,8]. Furthermore, pulse wave velocity of the brachial artery (PWVba) is a recognized marker of arterial stiffness in adults. Although different values have been proposed for cardiovascular risk identification (e.g., PWVba > 18 m·s^−1^ [9]), the >10 (m·s^−1^) PWVba value is well accepted as an indicator of high cardiovascular risk [10]. On the other hand, carotid-intima media thickness (cIMT) is a well-established marker of “structural” vascular health [11]. Despite this, there is a scarcity of proposals and clear agreement about cut-off points for considering high cardiovascular risk in adults. Values of cIMT > 0.9 mm have been suggested by expert panels as part of the proposals for considering high cardiovascular risk [10]. 

Physical inactivity, which refers to not following the international physical activity guidelines of 300 min per week of low-moderate physical activity or at least 150 min per week of vigorous-intensity physical activity [11,12], is more prevalent in sedentary, obese, and hypertensive populations, as well as those with dyslipidemia or metabolic syndrome, and is associated with negative effects on both functional and structural vascular parameters, such as flow-mediated dilation (FMD), pulse wave velocity of the brachial artery (PWVba), and carotid intima-media thickness (cIMT) [10,12,13,14]. Several expert panels have recommended moderate-intensity continuous training (MICT) for 30–60 min per session most days of the week for individuals with elevated blood pressure or hypertension [15]. MICT has been shown to be crucial for preventing and treating hypertension [16,17], and recent evidence has highlighted the time-efficiency of high-intensity interval training [18,19] and resistance training for improving EDys in a similar manner [20]. 

However, before starting any exercise training program in clinical populations such as those with elevated blood pressure or HTN, it will necessary to know the baseline cardiovascular response to physical effort through a progressive exercise volitional test, such as a cycling test [16,17]. The Astrand test is a useful progressive volitional cycling test that provides information about cycling power output in watts, which increases at different levels. For example, in women, power output increases by 25 watts per level, while in men, it increases by 50 watts. Heart rate should also increase progressively at each level. [17,18]. Interestingly, the theoretically predicted heart rate maximum (HR_predicted_) using the well-known formula (i.e., 220-age) is often overestimated or underestimated in physically inactive individuals [19]. Additionally, the use of heart rate maximum (HR_max_) is poorly reported in physically inactive hypertensive populations who are generally unable to maintain a steady state at maximal intensity. Therefore, the heart rate peak (HR_peak_) use, is a more easy and useful cardiovascular marker to obtain under exercise test conditions in physically inactive populations and has been widely reported for exercise prescription. This aim of this study was to assess the association between the EDys markers FMD, PWVba, and cIMT with the heart rate during a cycling test in HTN adults. A secondary aim was to characterize cardiovascular, anthropometric, and body composition outcomes in this population.

## 2. Materials and Methods

### 2.1. Participants

This preliminary descriptive study is part of an experimental randomized controlled clinical trial in which 75 adult men and women were invited to participate in an exercise training intervention and were assigned to one of three groups based on their blood pressure levels: arterial hypertension (HTN), elevated blood pressure (Ele), or a normotensive control group (CG). The study was conducted in Concepción, Chile between September 2022 and January 2023.

To determine the sample size, we used a G*Power 3.1.9.7 statistical sample size software calculator with an alpha error probability of *p* < 0.05 and a 95% confidence interval (CI) for three groups, expecting a medium-to-large effect size. Thus, a minimum of ten subjects per group would give a statistical power of ≥90%)]. 

The eligibility criteria for this study were as follows: (i) HTN, elevated blood pressure (controlled and on updated pharmacotherapy), or healthy normotensive; (ii) normal weight, overweight, or obese (as determined by body mass index [BMI]); (iii) normal or hyperglycaemic (T2DM, controlled and on updated pharmacotherapy); (iv) living in urban areas of the city of Concepción; and (v) the demonstrated ability to adhere to all measurements and stages of the study. Exclusion criteria included: (i) abnormal ECG; (ii) uncontrolled HTN (SBP ≥ 169 mmHg or DBP > 95 mmHg); (iii) morbid obesity (BMI ≥ 40 kg/m^2^); (iv) type 1 diabetes mellitus; (v) cardiovascular disease (e.g., coronary artery disease); (vi) diabetes complications such as varicose ulcers on the feet or legs, or a history of wounds, nephropathies, or muscle-skeletal disorders (e.g., osteoarthrosis); (vii) recent participation in weight loss treatment or exercise training programs (within the past 3 months); and (viii) the use of pharmacotherapy that can influence body composition.

All participants were informed about the study procedures and potential risks and benefits, and provided written consent. The study was conducted following the Declaration of Helsinki and was approved by the Ethics Committee of Universidad Andres Bello, Chile (Approval N° 026/2022). The clinical trial is registered under the clinical trials.gov international scientific platform under the code NCT05710653.

In the first stage of the enrolment (*n* = 75), subjects were screened, and after exclusion criteria (*n* = 10) participants were excluded for several reasons; ([*n* = 3] due to bone diseases, [*n* = 3] due to a history of heart disease, [*n* = 3] because they were already enrolled in other exercise activities, and (*n* = 1) due to be under weight loss treatment). Thus, a total of (*n* = 65) subjects participated in this first stage of our clinical trial study. The final sample size was as follows per group: (HTN *n* = 18, Ele *n* = 22, and CG *n* = 21). The study design can be seen in (Figure 1).

### 2.2. Endothelial Dysfunction Outcomes

To the three main EDys outcomes (FMD, PWVba, and cIMT), an ultrasound imaging 7–12 MHz linear-array transducer (GE Medical Systems, Model LOGIQ-E PRO, Milwaukee, WI, USA) for non-invasive vascular measurements of the brachial and carotid arteries was used. All participants were informed about refraining from eating, exercising, consuming caffeine, or taking vasoactive drugs before the test.

#### 2.2.1. Flow-Mediated Dilation

To measure FMD, each participant was positioned in a supine position and allowed to rest for 20 min. An ultrasound probe with a 60° inclination angle was then used in a longitudinal plane to explore the anterior and posterior lumen-intima interfaces at a site 1–3 cm proximal in the antecubital fossa to measure the brachial diameter and central flow velocity (pulsed Doppler) before the occlusion. The arm was abducted approximately 80° from the body and the forearm was supinated, and an adjustable mechanical metal arm precision holder with a magnetic base for a three-axis (X-Y-Z) positioning stage (EDI^TM^, Progetti e Sviluppo, Italy) was used to standardize the position and avoid evaluator bias. Next, a blood pressure cuff was positioned on the left arm and inflated at 50 mmHg (over the SBP baseline) for 5 min. Information was recorded during this time, including (i) a baseline image that was obtained before the occlusion, (ii) a 3-min video obtained (60 s before the stopping of the occlusion that was maintained until 2 min after cuff deflation), and (iii) a final image that was taken after the occlusion. The peak artery diameter after cuff deflation was recorded, and FMD was calculated as the percentage (%) rise in peak diameter from the preceding baseline diameter and the image after deflation [21], using the following formula: $$\mathrm{FMD}\;\left(\%\right)=\frac{\left[\left(\mathrm{peak}\;\mathrm{diameter}\;-\mathrm{baseline}\;\mathrm{diameter}\right)\right]*100}{\mathrm{baseline}\;\mathrm{diameter}}$$

The intra-session coefficient of variation has been ≤1% for the baseline diameter in our previous studies [18]. Reliability was estimated by intra-class correlation coefficients (ICC) based on four baseline measurements, with ICC values of 0.91 for the baseline diameter and ICC of 0.83 for FMD% (previously used data). 

#### 2.2.2. Carotid Intima-Media Thickness

To assess cIMT, we used the same ultrasound Doppler with the 7−12 MHz linear-array transducer. The participants were instructed to lie in a supine position and turn their heads slightly to the left and right. Once the carotid bulb was identified, a B-mode image was obtained for longitudinal right orientation of the common carotid artery. The scan was focused on 1 cm far from the bifurcation on the far wall of the common carotid artery. All images were recorded and analyzed offline using ultrasound software. Measurements were recorded at the end-diastolic stage, and the value for each side was obtained from the mean of three wall measurements of the cIMT [22]. A cIMT value of ≥0.9 mm was considered pathological [10], and a maximum thickness of ≥1.2 cm was indicative of pathological atherosclerosis [23]. 

#### 2.2.3. Pulse Wave Velocity

The PWVba was measured by analyzing oscillometric pressure curves that were registered from the upper arm in the brachial artery, and the measurement was represented in (m·s^−1^). An electronic device with a cuff for inflation/deflation positioned on the left arm (Arteriograph, TENSIOMED^TM,^ HU) was used for the measurement. This equipment automatically inflates/deflates the cuff and maintains occlusion in the left arm for 5 min to complete a pre-test/post 5-min test occlusion. After the measurement, the information was analyzed by a computer program (Arteriograph Software v.1.9.9.2; TensioMed, Budapest, Hungary) and a PDF information sheet was downloaded. The algorithm used to measure blood pressure in the Arteriograph^TM^ device had been previously validated [24]. A cut-off point of PWVba > 10 (m·s^−1^) denotes a high arterial stiffness risk, and thus a high cardiovascular risk [10]. An example representation of FMD, PWVba, and cIMT measurements can be seen in (Figure 2).

#### 2.2.4. Blood Pressure and Heart Rate at Rest

In a seated position and with at least 10 min of rest, systolic (SBP) and diastolic blood pressure (DBP) were classified by arterial hypertension (HTN), elevated blood pressure (Ele), or normotensive control condition (CG) following the last American Heart Association categorization (i.e., ‘normal’ blood pressure SBP/DBP <120/<80 mmHg, ‘Elevated’ blood pressure SBP/DBP 120–129/<80 mmHg, ‘stage 1′ HTN 130–139/80–89 mmHg, ‘stage 2′ HTN SBP/DBP ≥140/≥90 mmHg) [25]. Measurements were performed with an OMRON^TM^ digital electronic BP monitor (model HEM 7114, Chicago, IL, USA). Two recordings were made using the electronic device with a cuff for inflation/deflation positioned on the left arm. Immediately after the BP measurement, each subject was provided with a heart rate watch monitor in the left hand (Model A370, Polar^TM^, Kempele, Finland), where the heart rate at rest was registered.

#### 2.2.5. Progressive Volitional Cycling Test and Heart Rate during Exercise

The modified Astrand progressive volitional cycling test was used to determine both heart rate and power output in watts in each HTN, Ele, and CG participant [26,27]. During the test, the heart rate was measured in the first (HR_25–50_), second (HR_50–100_), third (HR_75–150_), fourth (HR_100–200_), and fifth (HR_125–250_) stages of the test progression, with different load graduations for men and women. Considering the evident differences in cycling performance from our HTN, Ele, and CG, some individuals will perform more than others in the test, thus, to ensure more robust statistical analyses with our associative regression models, we only included the first three stages of the Astrand test, particularly due to all subjects adhering to the completion of these stages. For the test, an electromagnetic cycle ergometer (model Ergoselect 200, ERGOLINE^TM^, Lindenstrasse, Germany) was used. The heart rate was continuously monitored using a telemetric heart rate sensor (Model A370, Polar^TM^, Finland), where we registered the maximum heart rate of each Astrand test stage.

We used the modified Astrand progressive volitional cycling test to measure both heart rate and power output in watts for each participant in the HTN, Ele, and CG groups [26,27]. The test involved measuring the heart rate in five different stages of test progression, with different load graduations for men and women: the first (HR_25–50_), second (HR_50–100_), third (HR_75–150_), fourth (HR_100–200_), and fifth (HR_125–250_) stages. However, due to differences in cycling performance among participants, we only included the first three stages of the test in our statistical analyses in order to ensure more robust results in our associative regression models. We used an electromagnetic cycle ergometer (model Ergoselect 200, ERGOLINETM, Germany) to conduct the test and continuously monitored the heart rate using a telemetric heart rate sensor (Model A370, Polar^TM^, Finland), recording the maximum heart rate for each stage.

#### 2.2.6. Anthropometric and Body Composition (Secondary Outcomes)

We measured body mass (kg), waist circumference (cm), body fat (%, kg), and skeletal muscle mass (%), as well as height (m). The first four variables were measured using a digital bio-impedance scale (OMRON^TM^ model HEM 7114^TM^, Chicago, IL, USA), while height was measured using a stadiometer (SECA^TM^, Model 214, Hamburg, Germany). Participants wore light clothing and no shoes during the measurements. We calculated body mass index (BMI) using body mass and height measurements to determine the degree of obesity according to standard criteria for normoweight, overweight, or obesity. We also recorded the basal metabolic rate and estimated body age. Table 1 presents the baseline characteristics of the study groups.

### 2.3. Statistical Analyses

Data are presented as mean with standard deviation (±SD). The Shapiro-Wilk test was used to test the normality assumption of all variables. The Wilcoxon rank sum test was used for variables that were not normally distributed. A one-way ANOVA was performed to test differences between groups, adjusted for weight, height, gender, SBP, and the use of beta-blockers. Additionally, a post-hoc Tukey’s test was applied after the ANOVA for group comparisons (HTN × Ele × CG). We also reported a trend analysis (ptrend) to test for potential (linear) tendencies to increase or decrease a particular outcome through the categories of different blood pressures. These analyses were applied using the Graph Pad Prism 8.0 software (Graph Pad Software, San Diego, CA, USA).

Finally, linear regression was applied to associate EDys outcomes (FMD, PWVba, cIMT) with heart rate (beats/min) in the first three steps of the progressive volitional cycling Astrand test (i.e., 25/50, 50/100, and 75/150 watt). The βeta value (for association), and R^2^ (for predictive capacity) were tested with heart rate for these EDys outcomes. In the regression model, each HR_25–50_, HR_50–100_, and HR_75–125_ watt was used as an independent model predictor of FMD, PWVba, and cIMT (in backward manner) adjusted for weight, height, gender, and SBP. These statistical analyses were performed with SPSS statistical software version 18 (SPSS™ Inc., Chicago, IL, USA), and statistical significance was set at *p* ≤ 0.05.

## 3. Results

### 3.1. Baseline Characteristics

As was the nature of the study, there were higher significant values of blood pressure in SBP comparing HTN vs. CG (143.2 ± 9.1 vs. 110.4 ± 7.0, *p* < 0.0001), and Ele vs. CG (124.9 ± 2.6 vs. 110.4 ± 7.0, *p* < 0.001) (Table 1). Similar results were shown in DBP comparing HTN to. CG (87.3 ± 10.7 vs. 73.8 ± 7.2, *p* < 0.0001), and Ele to. CG (83.3 ± 7.9 vs. 73.8 ± 7.2, *p* < 0.001) (Table 1), in MAP comparing HTN to CG (105.9 ± 10.2 vs. 86.0 ± 7.1, *p* < 0.0001), and Ele to CG (97.1 ± 6.1 vs 86.0 ± 7.1, *p* < 0.001) (Table 1), and PP comparing HTN to CG (55.9 ± 1.6 vs. 36.6 ± 2.2, *p* < 0.0001), and Ele to CG (41.6 ± 5.3 vs. 36.6 ± 2.2, *p* < 0.001) (Table 1).

### 3.2. Anthropometric and Body Composition (Secondary Outcomes)

In BMI, there were significant differences between HTN vs. the Ele group (29.5 ± 4.7 vs. 29.7 ± 3.8 kg/m^2^, *p* < 0.001), and between Ele vs. the CG group (29.7 ± 3.8 vs. 26.2 ± 3.1 kg/m^2^, *p* < 0.001). There was a significant trend (*p* = 0.004) to increase BMI from the CG to the Ele and HTN groups (Figure 3A). In waist circumference, there were significant differences between HTN vs. the Ele group (99.8 ± 8.7 vs. 100.1 ± 10.1 cm, *p* < 0.001), and between the Ele vs. CG group (100.1 ± 10.1 vs. 90.2 ± 10.5 cm, *p* < 0.001) (Figure 3B). There was a significant trend (*p* = 0.001) to increase waist circumference from the CG to the Ele and HTN groups (Figure 3B). There were no differences among groups in terms of body fat (%), skeletal muscle mass, or body age.

### 3.3. Endothelial Dysfunction Parameters (Main Outcomes)

In FMD, PWVba, and cIMT there were no significant differences among groups (Figure 4A–C). There was a significant trend (*p* = 0.047) to increase PWVba from the CG to the Ele and HTN group (Figure 4B). In cIMT, there were no significant differences among groups (Figure 4C). 

### 3.4. Heart Rate during Progressive Volitional Cycling Test in the HTN, Ele, and Control Groups

The description of heart rate during the progressive volitional cycling test is shown in (Figure 5). In the HTN group, the HR_predicted_ was of 177.7 beats/min, while the HR_peak_ in the cycling test was of 166.0 beats/min (Figure 5A). In the Ele group, the HR_predicted_ was 181.6 beats/min, while the HR_peak_ in the cycling test was 163.5 beats/min (Figure 5B). In the CG group, the HR_predicted_ was 180.0 beats/min, while the HR_peak_ in the cycling test was 152.8 beats/min (Figure 5C). Overall, each HTN, Ele, and CG normotensive group had a progressively increased heart rate from each cycling stage of 25–50 w, 50–100 w, 75–150 w, 100–200 w, and 125–250 w, from 96.8 to 166.0 in HTN (+69.2 beats/min), from 93.8 to 163.5 in Ele (+69.7 beats/min), and from 907 to 152.8 in CG (+62.1 beats/min) (Figure 5A–C). There were no significant differences in the HRR, HR_predicted_, and HR_peak_ among groups (Figure 5D–F). The HR_peak_ showed a significant increasing trend from CG (146.4) to the Ele (156.2), and the HTN group (159.5 beats/min) (Figure 5F).

### 3.5. Association between EDys Outcomes FMD, PWVba, and cIMT with Different Heart Rate during a Progressive Volitional Cycling Test in HTN, Ele, and Control Normotensive Subjects

When FMD and PWVba were correlated with the HR_25–50_, HR_50–100_, and HR_75–150_ watts of power output in the progressive volitional cycling test, no significant correlations were found in each HTN, Ele, and CG group (Figure 6A–F). Similarly, no significant correlations were found between cIMT with the HR_25–50_, HR_50–100_ steps of the Astrand test (Figure 6G,H). Although cIMT do not show an association with HR_75–150_ in the CG and the Ele group, there was a significant correlation between cIMT with HR_75–150_ in the HTN group (R^2^ 47.1, β −0.650, *p* = 0.038) (Figure 6I).

## 4. Discussion

The study found that there was a significant association between the vascular outcome cIMT and heart rate during the third stage of the cycling test in individuals with HTN (i.e., HR_75–150_). The heart rate during different stages of the test had a high predictive range for EDys outcomes, including FMD, PWVba, and cIMT in the HTN group. Additionally, there was a trend towards increased cIMT and PWVba in individuals with HTN compared to those with normal blood pressure, but no differences were observed in FMD. These findings were observed in HTN patients who had a higher prevalence of overweight/obesity as indicated by weight, BMI, and WC measurements.

Participants with HTN showed a significant association between the vascular outcome cIMT and heart rate during the third stage of the cycling test. The heart rate during different stages of the test had a high predictive range for EDys outcomes, including FMD, PWVba, and cIMT in the HTN group. 

Previous studies have reported a direct relationship between cIMT and an attenuated chronotropic response in a stress test, as was observed in this study [28]. It seems that the link between cIMT and the attenuated chronotropic response to exercise is the imbalance of the sympathovagal activation, which reflects the baroreflex sensitivity dysfunction. This dysfunction has been described as a probable cause of early atherosclerosis risk factors such as inflammation [29,30]. Therefore, an impaired chronotropic response to exercise could be an indicator of the EDys presence in subjects without cardiovascular risk factors, including HTN, as we found [28]. The study also observed that individuals with HTN had a higher prevalence of overweight/obesity as indicated by weight, BMI, and WC measurements. These findings emphasize the importance of monitoring body weight and body composition as part of the HTN management and EDys.

The findings are consistent with previous research indicating that individuals with HTN are more likely to be overweight or obese, as evidenced by measures of weight, body mass index (BMI), and waist circumference (WC) (Table 1). Furthermore, studies have shown that individuals with HTN and other cardiovascular risk factors, such as physical inactivity and a significant consumption of unhealthy foods tend to have impaired vasodilation after standardized FMD trials, high PWVba, and high cIMT [3,6]. 

The present study’s findings regarding PWVba are consistent with those of Park et al., who found that PWVba and FMD increased as frailty status increased in older adults. Specifically, PWVba increased from the non-frail group (1615.7 ± 209.9 cm/s [i.e., 16.1 m·s^−1^]) to the pre-frail (1815.2 ± 265.0 cm/s [i.e., 18.1 m·s^−1^) and frail (1829.9 ± 256.0 cm/s [i.e., 18.2 m·s^−1^) groups, which is similar to the increasing trend observed in our normotensive (7.7) to elevated (8.4) and hypertensive blood pressure group (8.7 m·s^−1^) (Figure 4). Furthermore, Park et al. found that FMD was lower in the pre-frail and frail groups (3.4% and 3.1%, respectively) compared to the non-frail group (5.2%), and with the frail group showing approximately two times lower FMD than the non-frail group [31]. In contrast, our study found that FMD was higher in the normotensive group (17.3%) compared to the HTN group (15.2%). However, this difference may be partially explained by the age difference between the two studies, as the average age of the groups in our study (HTN: 42.2 years, Ele: 38.3 years, CG: 39.9 years) was younger than the groups in the study by Park et al. (non-frail: 74.1 years, pre-frail: 75.3 years, frail: 75.3 years).

Our study found a significant association between the vascular outcome cIMT and heart rate during the third step of a cycling exercise test in individuals in the elevated blood pressure group (Figure 5). Additionally, heart rate during the different stages of the exercise test had a high predictive range for EDys outcomes, including FMD, PWVba, and cIMT. There was also a trend towards increased cIMT in individuals with elevated blood pressure compared to those with normal blood pressure, but no differences were observed in FMD or PWVba. Previous research has not found an association between resting heart rate and vascular parameters such as FMD. Our findings, which show an association between heart rate during exercise and EDys parameters, could be useful for predicting the behavior of vascular parameters and avoiding more invasive, expensive, and lengthy clinical tests.

In conclusion, this study provides evidence for the association between markers of endothelial dysfunction and heart rate during a progressive volitional cycling test in individuals with HTN. These findings suggest that monitoring heart rate during exercise testing could be a useful tool for assessing the risk of EDys in individuals with HTN. In addition, the study highlights the importance of monitoring body weight and composition as part of the management of HTN and EDys.

### Strengths and Limitations

As for limitations, our study had several constraints. Firstly, all variables were only measured in the afternoon, and PWVba was measured using an oscillometric cuff digital device instead of the more commonly used tonometry method. However, the equipment used for PWVba measurement has been validated previously. Secondly, we did not measure metabolic or plasma parameters, as they were not the primary objectives of this study. Nevertheless, none of the participants reported receiving hypercholesterolemia/dyslipidemia treatment within the past 6 months.

## 5. Conclusions

Heart rate during a progressive cycling exercise test is associated with vascular parameters in hypertensive patients, particularly during the second and third stages of the Astrand exercise test, indicating a high predictive capacity for vascular parameters in HTN when compared to control normotensive peers. Further research is needed to investigate the mechanisms behind these results.

## Figures and Tables

**Figure 1 ijerph-20-04236-f001:**
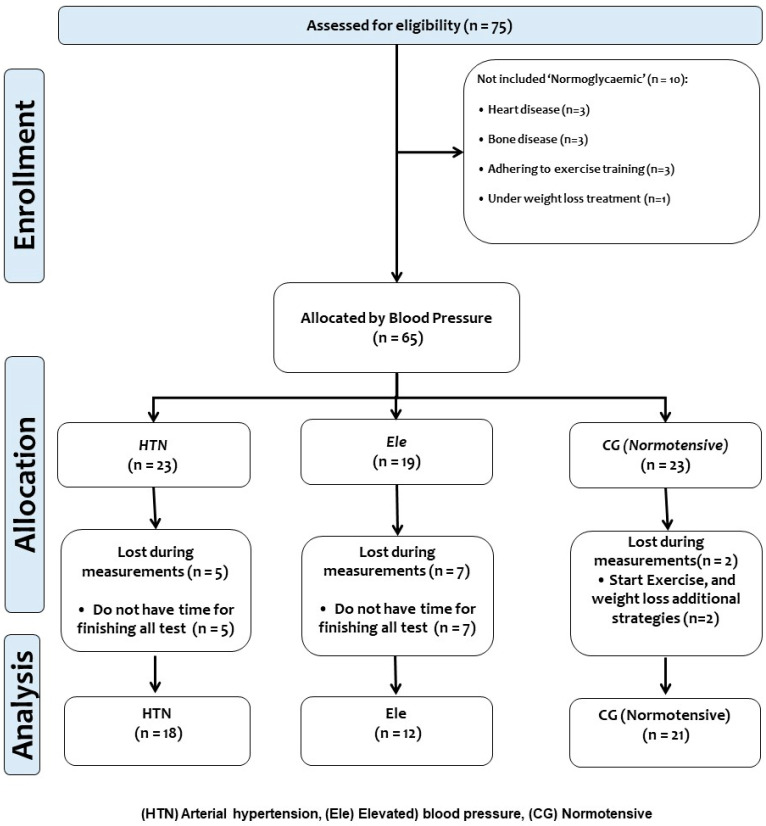
Study design.

**Figure 2 ijerph-20-04236-f002:**
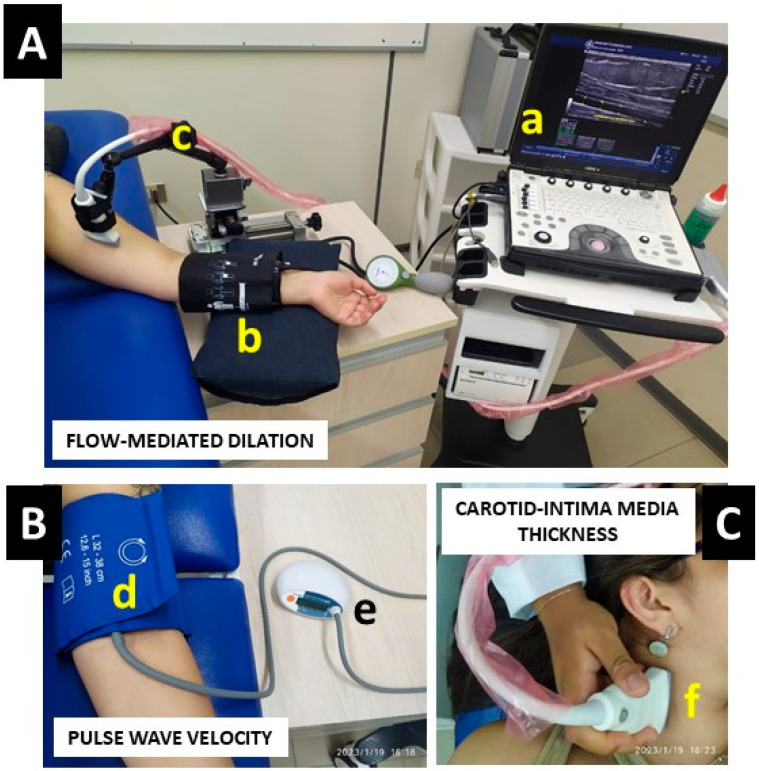
Flow-mediated dilation (panel **A**), pulse wave velocity (panel **B**), and carotid-intima media thickness (panel **C**). Equipment involved in the three measurements includes; (a) a Doppler ultrasound, (b) a blood pressure cuff to promote occlusion in the arm, (c) an adjustable mechanical three-axis precision holder, (d) a blood pressure cuff for multiple inflation/deflation, (e) Arteriograph^TM^ equipment, and (f) a linear transducer for the 7–12 MHz frequency.

**Figure 3 ijerph-20-04236-f003:**
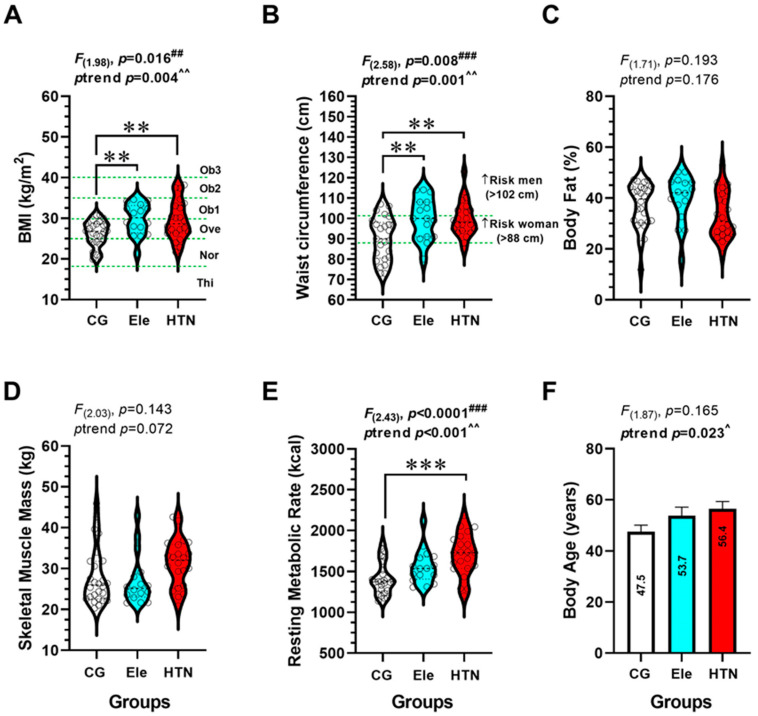
Anthropometric (body mass index (panel **A**), waist circumference (panel **B**)), and body composition outcomes (body fat % (panel **C**), skeletal muscle mass (panel **D**), resting metabolic rate (panel **E**)), and body age estimated (panel **F**) in three groups of adults with different blood pressure levels. The groups are described as: (CG) Control normotensive group, (Ele) Elevated blood pressure, and (HTN) arterial hypertension group. A green intermittent line denotes different adiposity and categorization cut-off points to BMI. Bold values denote significant modifications. (**) Denotes significant differences among groups at *p* < 0.001. (***) Denotes significant differences among groups at *p* < 0.0001. (##) Denotes interaction by ANOVA *p* < 0.001. (###) Denotes interaction by ANOVA *p* < 0.0001. (^) Denotes a significant trend across groups at *p* < 0.05. (^^) Denotes a significant trend across groups at *p* < 0.001.

**Figure 4 ijerph-20-04236-f004:**
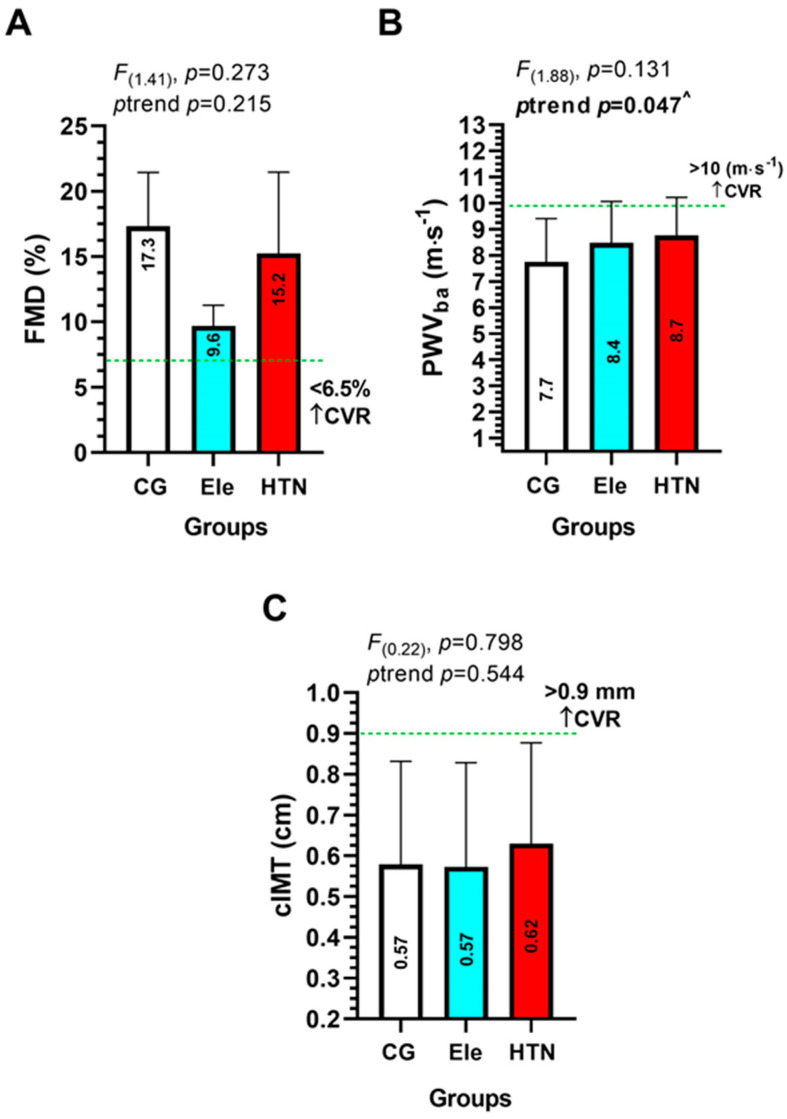
Endothelial dysfunction outcomes (flow-mediated dilation in % (panel **A**), aortic pulse wave velocity (panel **B**)), and carotid-intima media thickness (panel **C**)) in three groups of adults with different blood pressure levels. Groups are described as: (CG) Control normotensive group, (Ele) Elevated blood pressure, and (HTN) arterial hypertension group. (FMD) Flow-mediated dilation in %. (PWVba) Pulse wave velocity of the brachial artery, and (cIMT) Carotid intima-media thickness. A green intermittent line denotes different adiposity and categorization cut-off points to FMD, PWVba, and cIMT. (CVR) Denotes cardiovascular risk limit. (^) Denotes a significant trend across groups at *p* < 0.05.

**Figure 5 ijerph-20-04236-f005:**
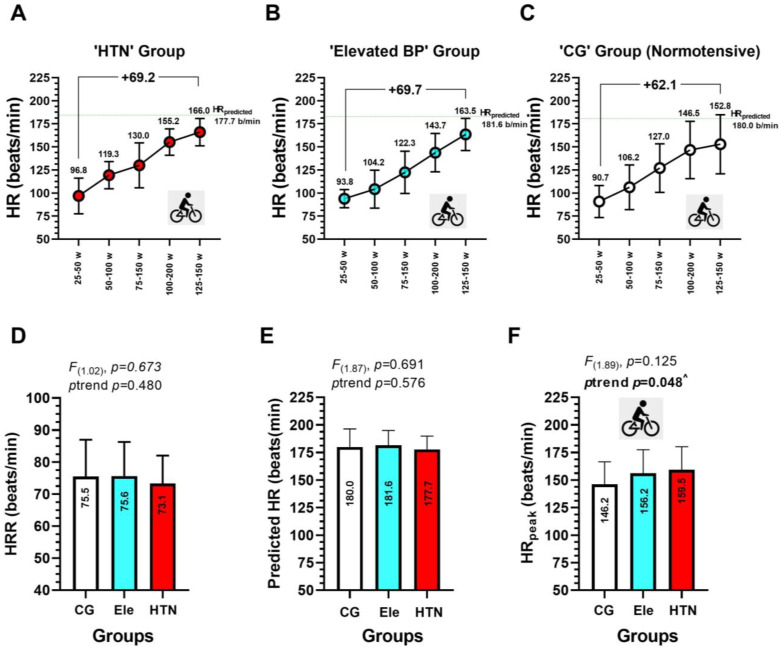
Heart rate outcomes during a progressive volitional cycling test in arterial hypertension (panel **A**), elevated blood pressure (panel **B**), and at the normotensive state (panel **C**). Heart rate at rest is shown in (panel **D**), heart rate predicted (220-age) is shown in panel (**E**), and heart rate peak in shown in panel (**F**). ^ Denotes a significant trend across groups at *p* < 0.05.

**Figure 6 ijerph-20-04236-f006:**
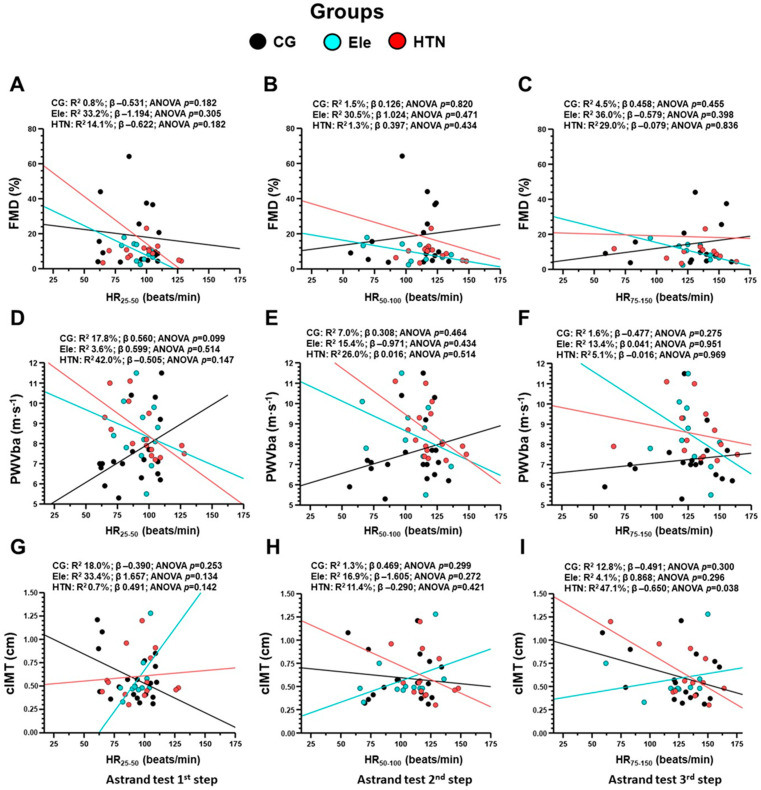
Association (by linear regression) between vascular parameters flow-mediated dilation (panels **A**–**C**), pulse wave velocity of the brachial artery (panels **D**–**F**), and carotid intima-media thickness (panels **G**–**I**) with heart rate at the first HR_25–50_ (panels **A**,**D**,**G**), second HR_50–100_ (panels **B**,**E**,**H**), and the third HR_75–150_ (panels **C**,**F**,**I**). Groups are described as: CG: Control normotensive group, Ele: Elevated blood pressure, and the HTN: arterial hypertension group. FMD: Flow-mediated dilation in %. PWVba: Pulse wave velocity of the brachial artery, and cIMT):Carotid intima-media thickness.

**Table 1 ijerph-20-04236-t001:** Baseline characteristics of three adult groups of different blood pressure control participants of a progressive volitional cycling test.

Outcomes	HTN	Ele	CG	Baseline *p*-Value
(n =)	18	12	21	
Anthropometric				
Age (y)	42.2 ± 12.2	38.3 ± 13.3	39.9 ± 16.2	*p* = 0.702
Weight (kg)	84.1 ± 15.1 *	79.7 ± 12.9 *	67.3 ± 10.5	*p* < 0.0001
Height (m)	1.70 ± 0.10 *	1.63 ± 0.07 *	1.59 ± 0.07	*p* < 0.0001
Blood pressure				
Systolic blood pressure (mmHg)	143.2 ± 9.1 ***	124.9 ± 2.6 **	110.4 ± 7.0	*p* < 0.0001
Diastolic blood pressure (mmHg)	87.3 ± 10.7 ***	83.3 ± 7.9 **	73.8 ± 7.2	*p* < 0.0001
Mean arterial pressure (mmHg)	105.9 ± 10.2 ***	97.1 ± 6.1 **	86.0 ± 7.1	*p* < 0.0001
Pulse pressure (mmHg)	55.9 ± 1.6 ***	41.6 ± 5.3 **	36.6 ± 2.2	*p* < 0.0001
Pharmacologic theatment				
Angiotensine converting enzyme inhibitors (n = /total)	2/18	-	-	
Βeta blocker (n = /total)	2/18	-	-	
Metformin (n = /total)	-	1/18		

Data are shown as mean and ± standard deviation. (*) Denotes significantly different versus CG at *p* < 0.05. (**) Denotes significantly different versus CG at *p* < 0.001. (***) Denotes significantly different versus CG at *p* < 0.0001.

## Data Availability

The datasets of the current study are available from the corresponding author on reasonable request.

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
