# Peer review of "Heart Rate from Progressive Volitional Cycling Test Is Associated with Endothelial Dysfunction Outcomes in Hypertensive Chilean Adults"

_ijerph, 2023, doi:10.3390/ijerph20054236_

Round 1

Reviewer 1 Report

1. This paper was intended to predict the endothelial dysfunction of hypertensive patients by using heart rate. This is very good idea. However, this paper has only done a correlation analysis which does not give adequate support to the article title. It is necessary to calculate a regression equation between HR and cIMT. In addition, "prediction" in the title means to speculate on a future pathological state in the current healthy state. But the endothelial function of the participants was not clearly described.

2. Line 76, “…where >10 cm/s indicates high cardiovascular risk”, please recheck this information. “As a result, the PWVba cut point for predicting incident hypertension was 1319 cm/s for male (sensitivity: 58.5%, specificity: 59.7%) and 1246 cm/s (sensitivity: 69.5%, specificity: 69.5%) for female.” This is one of the results from the paper you cited that may be useful to you. In addition, please confirm the unit of PWV in your paper is cm/s or m/s?

3. Line 123, please clarify why must participants live in urban areas of the city of Concepción?

3. Please clarify the definition of elevated blood pressure, such as the range of systolic and diastolic blood pressures.

4. Line 224-225. The measurement of the heart rate should be described in more detail. For example, the average or maximum HR for each stage? And, Figure 5, Panel D. When is the HR at rest measured? Why are the HRs at rest so high in all groups?

5. As was shown in line 224-225, there were 5 stages in test progression. Please clarify why correlation analysis was only carried out in 3 stages..

Author Response

Dear reviewer,

many thanks by the opportunity to improve our work. By the present we hope to be addressed with success all your comments.

Reviewer 2 Report

In this paper Cristian Alvarez and collegues investigate the relationship between endothelial dysfunction markers (FWD, PWVba and cIMT) and heart rate during the ASTRAND test.

While this is a well-presented study, several points needs to be addressed:

1) may the authors explain why participants with BMI>= 40 were excluded? This is a strong limitations considering their adiposopathy and the well-known  "vascular and metabolic dysfunction" burden;

2) may the author add to the baseline table, some important medical history informations like "cholesterols and triglycerides levels", "hypertension mediated organ damage" components, previous MACEs  and "pharmacological treatments"?;

3) line 355-358 "there was a significant trend towards increased cIMT in the HNT group compared to the normotensive control group and those with elevate blood pressure": if three different linear regression model was computed, Authors should revise as "there was a significant trend towards increased cIMT in the HNT group and no significant trend [...]  in the normotensive control group and those with elevate blood pressure;

4) may the authors provide more information of the linear regression models of Figure 5 (adjusted R squared, R, standard error, etc)?

Author Response

(The authors gave the same response as above.)

Round 2

Reviewer 2 Report

Dear Alvarez and collegues, prior to the publication of your work, I have another two suggestion:

1) as Authors state on the coverletter, this study focus on a population that is representative of Chilean participant. On the basis of that, I suggest to adapt the title of the Study as follows: [...] in Hypertensive Chilean Adults". 

2) ANOVA analysis should be adjusted for the major confounding variable that is the use of betablocker;

Author Response

Dear reviewer,

Our responses are into the file. Many thanks
